# Contrasting cognitive, behavioral, and physiological responses to breathwork vs. naturalistic stimuli in reflective chamber and VR headset environments

**Ninette Simonian**[1], **Micah Alan Johnson**[1,2], **Caitlin Lynch**[1], **Geena Wang**[1], **Velu Kumaravel**[1], **Taylor Kuhn**[3], **Félix Schoeller**[1], **Nicco Reggente**[1]*

**1** Institute for Advanced Consciousness Studies, Santa Monica, California, United States of America,
**2** Department of Biostatistics, Johns Hopkins University, Baltimore, Maryland, United States of America,
**3** Department of Psychiatry and Behavioral Sciences, University of California, Los Angeles, California, United States of America

* nicco@advancedconsciousness.org

## Abstract

The MindGym, a novel immersive technology utilizing a reflective chamber environment, was developed to create standardized experiential content, including anxiolytic experiences. This study examined whether therapeutic experiences originally created in the MindGym could maintain their efficacy when delivered via 360-degree recordings through virtual reality (VR) headsets. A randomized controlled trial (N=126) compared anxiety reduction, cognitive performance, and physiological responses across four conditions: MindGym and VR platforms, each delivering either breathwork or rain stimuli. Results demonstrated significant improvements across all conditions in cognitive performance (Trail Making Test RTACC, p.fdr<.001; Architex Total Speed, p.fdr<.001) and anxiety reduction (STAI, p.fdr<.001). Breathwork conditions produced greater decreases in breath rate compared to rain stimuli (p.fdr=.002). Treatment responses were moderated by individual differences, with absorption (MODTAS) predicting both awe (p.fdr=.004) and ego dissolution (p.fdr=.015), while openness to experience interacted with stimuli type to influence anxiety reduction (p.fdr=.038). The anxiolytic effects originally generated in the MindGym maintained full efficacy when translated to VR delivery, with no significant differences in effectiveness or immersion between the original environment and its virtual reproduction. These findings establish the MindGym as a viable content creation platform for immersive, anxiety-reducing experiences that can be successfully adapted to more accessible delivery systems, while highlighting the potential for personalization based on individual differences. Future research should investigate the translation of more complex MindGym-generated experiences to expand accessible anxiety management tools.

## Introduction

Experience shapes human cognition and behavior, yet its variability limits prescriptive outcomes. Experiential technology, leveraging audiovisual stimuli, offers potential for targeted

**Data availability statement:** All raw data has been uploaded to an OSF repository for this paper with the following link: https://osf.io/kch59/?view_only=577b4ece1b2b47be-8f4a6f063f9e8744.

**Funding:** This research was financially supported under a Research Services Agreement executed between the Institute for Advanced Consciousness Studies (a 501(c)(3) non-profit) and Lumena, Inc. (Denver, CO). It is imperative to note that the terms of this agreement did not incorporate any provisions or incentives contingent upon the attainment of specific outcomes or successful results; the agreement was solely predicated on underwriting the requisite expenses for the study's execution.

**Competing interests:** This research was financially supported under a Research Services agreement executed between IACS and Lumena, Inc. It is imperative to note that the terms of this agreement did not incorporate any provisions or incentives contingent upon the attainment of specific outcomes or successful results but were solely predicated on underwriting the requisite expenses for the genuine and unbiased execution of the study. Lumena, Inc. had no input on the design nor analysis of this study, apart from supplying MindGym's content, delivery system hardware and software, and the 360 videos utilized in the VR.

interventions with predictable effects on mood and cognition [1–4]. Immersive virtual reality (VR) experiences can elicit specific emotional responses (e.g., awe) and enhance well-being through increased felt connectedness [5].

Anxiety disorders, affecting 19.1% of adults globally, significantly impact healthcare systems and individual well-being, with even subclinical levels diminishing quality of life [6–8]. First-line treatments like cognitive-behavioral therapy and medication offer benefits but face limitations such as side effects and relapse potential [9,10]. Alternative approaches (mindfulness, exercise, pharmacotherapies) show variable efficacy [6,10,11], while emerging interventions (vagal nerve stimulation, cannabidiol) require further safety and efficacy research [11,12]. This landscape necessitates innovative approaches to anxiety management.

Two promising avenues for acute anxiety reduction are exposure to naturalistic stimuli and guided breathwork. Nature exposure and related therapies have been shown to facilitate stress recovery, mood enhancement, improved cognitive functions, and overall well-being while alleviating anxiety symptoms and boosting health and self-efficacy [13–21]. Breathwork techniques demonstrate comparable efficacy through distinct mechanisms, inducing parasympathetic states that promote stress reduction and enhance emotional regulation [22–28].

Despite their potential, these approaches face accessibility challenges, particularly in urban environments with limited access to natural settings [29] and due to the lack of integration of breathwork and somatic therapies into standard mental health care [11]. While immersive technologies offer a solution by creating scalable, accessible virtual experiences, enabling well-controlled, "plug-and-play" interventions with optimized effectiveness and expanded reach. Innovative platforms for generating immersive content, such as the MindGym, hold promise for creating highly effective experiences that could potentially be adapted to various delivery systems, including more portable VR formats. Research shows combined audiovisual nature scenes offer enhanced psychological benefits over single-modality presentations [1,3], aligning with Attention Restoration and Stress Reduction theories [13,30]. 360-degree nature videos demonstrate rapid mood improvements [21,31–36], while natural sounds enhance physiological recovery [37] and cognitive performance [38].

The efficacy of these virtual interventions hinges on immersion—the technological capability to create a sense of "being there"—and presence, the subjective experience of feeling transported to the virtual environment. These factors are crucial for eliciting powerful psychological and physiological responses [39,40]. Crucially, research shows the cognitive benefits of VR depend on users truly feeling "there," emphasizing the fundamental role of high immersion and presence [41]. Both technological capabilities and subjective experiences like perceived realism and enjoyment influence these factors [42–44]. Immersion levels vary across VR systems, from moderate levels in consumer-grade headsets to heightened experiences in advanced setups [45,46]. These findings underscore the potential of immersive technology to create targeted, accessible interventions that leverage nature's restorative properties.

The current study investigated whether effective experiences created in one immersive technology could be successfully translated to another by comparing two immersive technologies in delivering anxiolytic content: a "reflective chamber" (MindGym) and a traditional VR headset. Both incorporated naturalistic rain sounds or guided 4-7-8 breathwork [24,47,48]. Our primary aim was to determine whether the experience designed for the MindGym could maintain their efficacy when adapted to more widely accessible VR platforms. We hypothesized that MindGym, designed for enhanced immersion and presence, would outperform VR by amplifying nature and breathwork effects while minimizing VR-associated discomfort [49–52]. This comparison sought to inform the development of novel, scalable anxiety-reducing interventions, bridging the gap between traditional therapies and the need for accessible mental health solutions.

## Methods

### Participants

A total of 126 participants (63 females; age range 18-74 years; mean 41, SD 15.19) participated in the study. Participants were randomly assigned to one of four groups, as shown in Table 1.

10 participants were excluded from the physiological analysis due to equipment failure or excessive movement resulting in a total of 116 participants for statistical analysis (61 females; age range 18-74 years; mean 40.54, SD 14.56). Respiration was analyzed for 88 participants, ECG was analyzed for 80 participants, and EDA was analyzed for 96 participants (see Table 2). All participants passed quality assurance checks, with respiration rates below 20 breaths per minute following data cleaning and processing (see Methods for further explanation).

Of the 126 enrolled participants, 87 completed a one-week follow-up survey (See Table 3). The similar distribution of demographic characteristics suggests random attrition, though this conclusion is limited by incomplete demographic data across the full sample.

### Recruitment

Participants were recruited for this study through newsletters sent to individuals with prior engagement in our previous research studies and targeted advertising on Facebook and

**Table 1. Participant characteristics by group and analysis type.**

| Group | N | Gender (M/F) | Age Range | M | SD |
|---|---|---|---|---|---|
| VR (Breathwork) | 33 | 16/17 | 20-71 | 42.61 | 15.18 |
| VR (Rain) | 29 | 16/13 | 18-69 | 41.30 | 15.54 |
| MindGym (Breathwork) | 31 | 15/16 | 22-74 | 38.45 | 15.93 |
| MindGym (Rain) | 32 | 16/16 | 19-68 | 41.53 | 14.50 |

F = female; M = male. Age is reported in years. M and SD represent the mean and standard deviation of age, respectively.

**Table 2. Participant characteristics for physiological measures by group.**

| Measure | Group | N | Gender (F/M) | Age Range | M | SD |
|---|---|---|---|---|---|---|
| **Respiration** | | | | | | |
| | VR (Breathwork) | 22 | 12/10 | 20-65 | 39.91 | 14.36 |
| | VR (Rain) | 19 | 8/11 | 18-69 | 43.15 | 16.56 |
| | MindGym (Breathwork) | 28 | 14/14 | 23-74 | 36.96 | 15.17 |
| | MindGym (Rain) | 19 | 11/8 | 19-63 | 41.89 | 14.11 |
| **ECG** | | | | | | |
| | VR (Breathwork) | 18 | 9/9 | 20-65 | 41.17 | 14.31 |
| | VR (Rain) | 16 | 9/7 | 18-68 | 41.44 | 17.00 |
| | MindGym (Breathwork) | 26 | 15/11 | 23-74 | 37.04 | 15.84 |
| | MindGym (Rain) | 20 | 12/8 | 19-68 | 41.80 | 13.33 |
| **EDA** | | | | | | |
| | VR (Breathwork) | 23 | 14/9 | 20-65 | 41.61 | 13.26 |
| | VR (Rain) | 22 | 9/13 | 18-69 | 40.77 | 15.72 |
| | MindGym (Breathwork) | 28 | 14/14 | 23-68 | 36.82 | 14.65 |
| | MindGym (Rain) | 23 | 14/9 | 20-63 | 41.43 | 13.03 |

F = female; M = male. Age is reported in years. M and SD represent the mean and standard deviation of age, respectively. ECG = Electrocardiogram; EDA = Electrodermal Activity.

**Table 3. Study participants by follow-up completion status across groups and demographics.**

| | | Completers (N=49) | Non-completers (N=24) |
|---|---|---|---|
| **Group** | | | |
| | VR (Breathwork) | 12 | 9 |
| | VR (Rain) | 9 | 6 |
| | MindGym (Breathwork) | 13 | 5 |
| | MindGym (Rain) | 15 | 4 |
| **Gender** | | | |
| | Male | 25 | 10 |
| | Female | 24 | 14 |
| **Education** | | | |
| | Bachelor's degree | 27 | 13 |
| | Master's degree | 9 | 3 |
| | Some college | 6 | 5 |
| | High school diploma | 5 | 1 |
| | GED or Equivalent | 0 | 1 |
| | Professional degree | 1 | 0 |
| | Associate's degree | 1 | 1 |
| **Race** | | | |
| | White | 23 | 11 |
| | Asian | 10 | 9 |
| | Multiracial | 6 | 3 |
| | Black/African American | 2 | 1 |
| | Hispanic/Latino | 4 | 0 |
| | Indigenous American | 1 | 0 |
| | Not disclosed | 3 | 0 |
| **Ethnicity** | | | |
| | Not Hispanic/Latinx | 39 | 23 |
| | Hispanic/Latinx | 9 | 1 |
| | Unknown | 1 | 0 |

Note: Due to a clerical error during data preparation, demographic data were available for only 73 of 126 enrolled participants (57.9%). Of the total enrolled sample, 87 participants (69.0%) completed the one-week follow-up survey. Similar distributions of demographic characteristics between completers and non-completers suggest random attrition, though this conclusion is limited by incomplete demographic data across the full sample.

Instagram, aimed at adults residing within a 50-mile radius of Santa Monica, California. Participants were compensated $30 per hour via cash or Venmo payments, and parking fees were validated. The total participation duration, measured from arrival to departure, was rounded up to the nearest 15-minute increment for compensation purposes. Recruitment began on 2023 April 24 and ended on 2024 March 22.

## Eligibility

Inclusion criteria for study participation were as follows: (1) absence of neurological conditions including epilepsy and migraine; (2) no history of claustrophobia; (3) absence of photosensitivity or ophthalmological conditions such as cataracts, corneal abrasions, keratitis, or uveitis; (4) normal or corrected-to-normal auditory and visual acuity; (5) no current use of medications known to induce photophobia or alter auditory perception; and (6) sufficient

mobility to enter the MindGym apparatus without assistive devices (e.g., wheelchairs, walkers, or canes).

Prior to enrollment, potential participants underwent screening via a standardized online questionnaire administered through the Google Forms platform. This pre-screening process ensured adherence to all inclusion and exclusion criteria before formal study initiation.

## Materials

All participants sat on an OMEGA Gaming Chair (SecretLab, Inc.) and completed the behavioral questionnaires (administered through Google Forms) and cognitive tasks on a 27" 2022 iMac (Apple, Inc.) regardless of group.

**Physiological monitoring.** The CGX Aim II Physiological Monitoring device (CGX, Inc.), using ECG electrodes (Skintact Inc.), measured the following bio peripherals: Electromyography (EMG; 2 electrodes on the L/R base of the neck on the sternocleidomastoid muscle), Bio-Impedance-Based Respiration Rate (2 paddles with 2 electrodes each on the L/R pectoralis major), Heart Rate, and Galvanic Skin Response (GSR; 2 electrodes on the palm of the non-dominant hand).

**Lab recorder.** Lab Streaming Layer with LabRecorder [6,10,11] was utilized to temporally synchronize our peripheral and experimental time series (experience start and end timestamps) within an XDF file format.

**Trait measures.**

a. Dispositional Positive Emotion Scale (D-PES;[53]) – The Awe Subscale of D-PES measures an individual's dispositional propensity to feel awe toward the world, consisting of 6 items.

b. Modified Tellegen Absorption Scale (MODTAS; [54]) – Adapted from the Tellegen Absorption Scale (Tellegen & Atkinson, 1974), uses a Likert scale to assess the imaginative involvement and the tendency to become mentally absorbed in everyday activities.

c. NEO Five-Factor Inventory (NEO-FFI-3; [55]) – Measures five broad personality dimensions – neuroticism, extraversion, openness to experience, agreeableness, and conscientiousness.

**State measures.**

a. State-Trait Anxiety Inventory (STAI;[56]) – Assesses state anxiety (temporary/situational) and trait anxiety (enduring propensity).

b. Profile of Mood States 2nd Edition (POMS 2; [57]) – Evaluates current mood state affective dimensions like anger, confusion, depression, fatigue, tension, and vigor.

c. Valence and Arousal – Measures emotional state on two Likert scales from 0 to 10: Valence (0= very unpleasant,10 = very pleasant) and Arousal (0 = totally calm, 10 = totally excited/agitated).

**Outcome measures.**

a. Toronto Mindfulness Scale (TMS; [58]) – A 13-item administered post-meditation to assess state mindfulness, while differentiating between reflective awareness and ruminative attention. The TMS contains two subscales: Curiosity, gauging the interest in one's experiences, and Decentering, reflecting the ability to view thoughts and feelings as transient mental events.

b. Awe Scale (Awe-S; [59]) – An 8-item self-report scale measuring individual differences in the tendency to experience awe, using a 7-point Likert scale to rate their agreement with statements regarding feelings of awe and perceptions of vastness.

c. Body Size Estimation (BSE; [60]) – Assesses body image perception, from a selection of nine progressive figural drawings that depict varying body sizes to estimate their own perceived body size, both in height and width. This measure has been linked to the intensity of awe and immersion in non-ordinary experiences [61].

d. Ego-Dissolution Inventory (EDI; [62]) – Consists of 16 items assessing altered states of ego-consciousness, using a visual analog scale ranging from 0% to 100%. It includes 8 items relating to ego-dissolution, representing the core of the inventory, and 8 to ego-inflation, contrasting experience of heightened self-assuredness.

e. Igroup Presence Questionnaire (IPQ; [63]) – Measures sense of presence, immersion, engagement, and connection in virtual environments.

f. Modified Immersion Experience Questionnaire (adapted per [64]) – Measures depth and quality of immersion/engagement in a particular context or virtual environment.

g. Motion Sickness Assessment Questionnaire [65] – A 16-item scale assessing multiple dimensions of motion sickness (gastrointestinal, central, peripheral, and sopite-related symptoms).

**Behavioral tasks.**

a. Trail-Making Test (TMT; [66]) – A neuropsychological assessment, taken on the computer, evaluating visual attention, scanning, processing speed, and cognitive flexibility. Participants were instructed to connect a series of 25 dots in sequential order, alternating between numbers and letters (1-A-2-B, and so on), aiming for speed and accuracy.

b. Architex [67] – A validated, computerized neuropsychological assessment tool designed to objectively measure multiple cognitive domains and their interactions, including attention, working memory, processing speed, affect recognition, planning, organization, impulsivity, and verbal and nonverbal abstract reasoning.

**Content delivery systems.**  The first immersive content-delivery system was MetaQuest 2 (Meta Platforms, Inc.) VR headset, equipped with an Elite Strap (see Fig 1A), presenting stimuli through its native video player at a resolution of 5376x2688, scaled to the maximum allowed resolution of 1920x3664, and a refresh rate of 120Hz. All videos were captured using a 360 camera (GoPro Max 360) placed inside the MindGym during the programmed experience, with the audio content subsequently overlaid. During the experiment, the VR is set to cast through an iPad Air (iPadOS 16.6.1) and navigated using a wired mouse (Razer Inc.).

The second system was the MindGym (Lumena, Inc.), a 7' isotropic MindGym lined with reflective mylar on its interior walls, while its floor and ceiling are equipped with mirrors. (see S1 Appendix). The MindGym incorporates WS2815 LEDs, with 121 pixels per edge between the vertices, totaling 1452 pixels across 12 edges. It utilizes SMD5050 RGB LED chip, offering RGB color with 256 grayscale levels and an output of 990-1080 lumens per meter. The LEDs are operated at less than half of their maximum capacity. Additionally, the system features a color temperature of 5500K. Participants in the MindGym group were provided over-ear wireless, noise-canceling headphones (Sony Group Corp.) to wear during the experience (see Fig 1B).

**Stimuli.**  The first piece of content was "4-8 Breathwork" (see Fig 2A, Fig 2C), led by Dr. Jannell MacAulay's guided audio and utilizing two primary lighting setups. The beginning

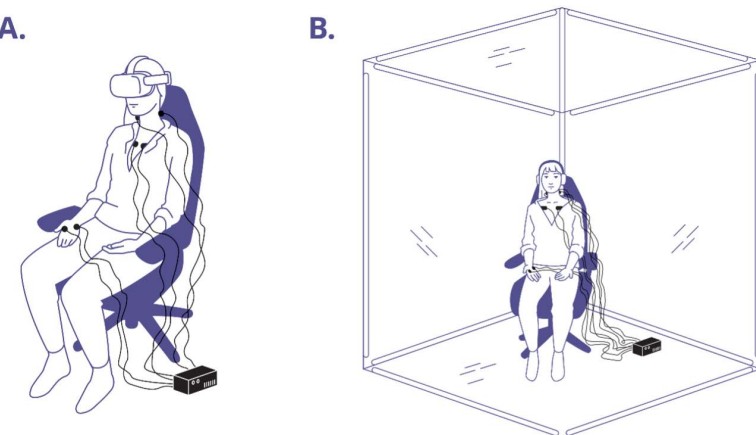

**Fig 1. Participant setup.** (A) VR; (B) MindGym.

featured the "Mindful Minute" mindfulness exercise with multiple colors and slowly upward-moving LEDs, designed to encourage mind wandering among participants. MacAulay then introduced the breathwork technique and its underlying rationale, aided by "Galaxy" visuals of blue LEDS, around 6000K, embedded in the walls, giving the sensation of being surrounded by stars. The guided breathwork audio was complemented by upward and downward cascades of vertically aligned LEDs, synchronized with the inhalation and exhalation periods (see S2 Appendix).

The second piece of content was "Rain" (see Fig 2B, Fig 2D), an immersive environment created for reflection. The audio begins with complete darkness as the sound of rain starts to trickle in. This is complemented by a downward cascade of vertically aligned blue LEDs, around 6500K, against a black background, simulating rain visuals falling from each vertex. The participant remains immersed in the light and sound environment for the remainder of the reflective experience.

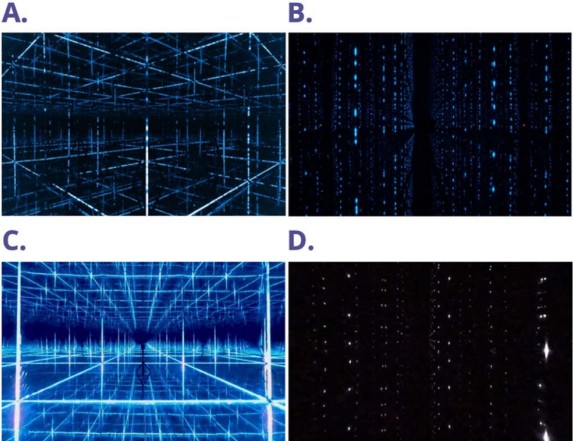

**Fig 2. Screenshot of stimuli.** (A) MindGym (Breathwork); (B) MindGym (Rain); (C) VR (Breathwork); (D) VR (Rain).

## Procedure

**Overview.** Before each session, all participants provided signed consent forms. They began the session by filling out pre-experience questionnaires and completing cognitive tasks, followed by CGX setup. Subsequently, each group underwent a 10-minute experiential phase, after which they completed post-experience questionnaires and tasks (see Fig 3).

**Randomization.** Subjects were iteratively assigned to one of two groups using a MATLAB-based randomization algorithm that maintained gender balance, with recruitment ceasing for each gender category upon reaching the predetermined quota. The experimenter remained blind to group assignment until launching the Python script, which initiated the group-appropriate content delivery system and stimuli by entering the participant's ID number. Participants were randomly assigned to one of four content delivery systems and stimuli pairings: (1) MindGym (Breathwork), (2) MindGym (Rain), (3) VR (Breathwork), and (4) VR (Rain).

**Procedure.** Upon arrival, participants were seated on an office chair facing a 35.5" x 55" desk and given an overview of the session (pre-questionnaires, equipment setup, 10-minute experience, post-questionnaires). They were informed their "experience" would involve sitting for 10 minutes in an immersive audiovisual environment.

For approximately the first 30 minutes, participants used the iMac to complete a set of questionnaires (NEO-FFI-3, D-PES, MODTAS, STAI, POMS-2, Valence, and Arousal) on Google Forms, followed by a brief neurophysiological assessment via the TMT and Architex. Participants were instructed to silence their phones and remove jewelry and bulky items for comfort, before being told which experiential group they were assigned to. Using a cotton pad and 91% isopropyl alcohol, the experimenter cleansed the relevant areas (see CGX materials above) before outfitting the participant with biosensors and peripherals.

Participants assigned to VR sat in the same room, with the room shades lowered. The VR headset was positioned and adjusted with the tension dial on the Elite strap for clear vision and comfort. The experimenter cast the VR to the iPad and adjusted the settings, such that the audio was played through the headset. Upon connecting to Lab Recorder, the stimuli are launched concurrently with the initiation of the LSL trigger, by the Python script.

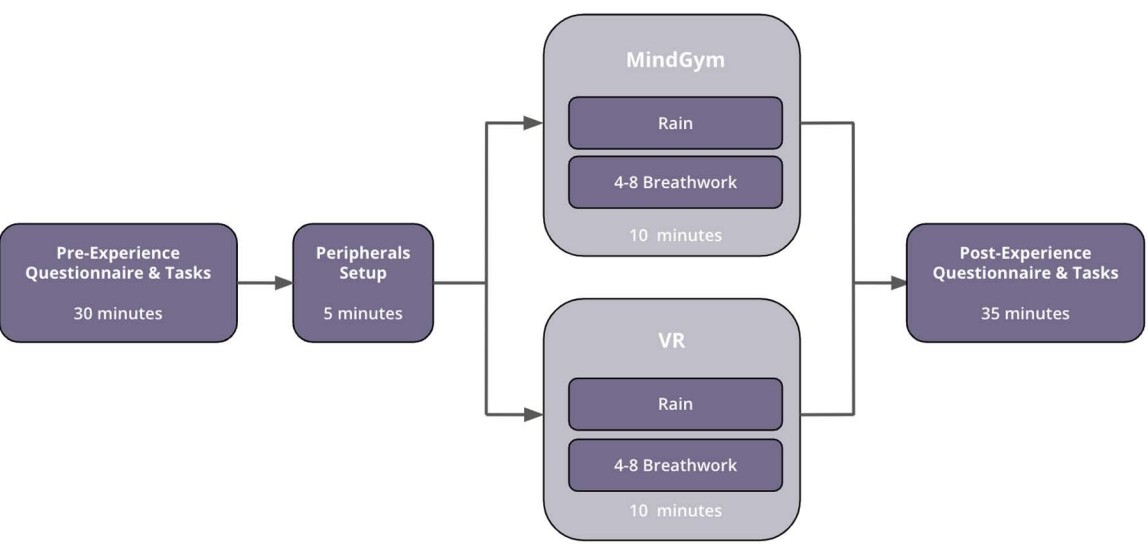

**Fig 3. Procedure overview.** The total participation time was roughly 80 minutes.

If assigned to the MindGym, participants entered the MindGym and sat centrally, while holding the CGX device and wearing headphones. The Lab Recorder was connected, and the experience began within 15 seconds of closing the MindGym door.

Across groups, participants were told to sit still and relax during the 10-minute experience without falling asleep. Participants were shown how to remove the VR or exit the MindGym if they decided to end the experience early, otherwise, the experimenter would return to assist them once the experience ended.

After the experience, biosensors were removed and participants used the restroom if needed before completing post-experience behavioral assessments (STAI, POMS-2, Valence, Arousal, IPQ, MSAQ, TMS, Awe-S, BSE, and EDI) and cognitive tasks (TMT and Architex). Finally, participants were compensated and dismissed.

## Behavioral statistical analysis

**Dependent variables.** Seven primary outcome measures collected at both timepoints were used as dependent variables (DV) in separate models. There were four mood state measures: Anxiety (STAI), Arousal, Valence, and Total Mood Disturbance (POMS). There were also three cognitive measures: two that tracked performance on the Architex task (Total Speed and Total Score) and one that tracked performance on the TMT using the ratio of reaction time divided by accuracy, here called TMT RTACC, which decreases with increasing performance due to increased accuracy and/or decreased reaction time).

Eleven DVs were collected at the Post timepoint only, which included Awe, BSE, EDI, Immersion (based on IEQ), overall Motion, Curiosity and Decentering (both from Toronto), and four IPQ variables (General, Involvement, Spatial Presence, Experienced Realism).

There were also four physiological variables – Breath Rate, EDA, HR, and HRV – that were collected throughout the duration of each experience. These measures were averaged or computed in either two 5-minute timepoints (for HRV), five 2-minute timepoints (for EDA and Breath Rate), or ten 1-minute timepoints (for HR).

**Independent variables.** The primary independent variables (IV) were Time or Timepoint (Pre, Post) as a categorical, within-subjects factor, for any variables collected at both timepoints, as well as both Tech (MindGym, VR) and Stim (Breathwork, Rain) as categorical, between-subjects group factors. Several continuous or ordinal IVs were tested for potential moderation or mediation of the Tech and Stim effects. There were eight psychological variables: DPES, Immersion, IPQ (General), IPQ (Involvement), IPQ (Spatial Presence), IPQ (Experienced Realism), MODTAS, and Openness. There were seven physiological variables: Motion (Overall), initial breath rate (Breath Rate, first 2 min), breath rate change (Breath Rate change, last 2 min – first 2 min), initial heart rate (HR, first 1 min), heart rate change (HR change, last 1 min – first 1 min), initial heart rate variability (HRV, first 5 min), and change in heart rate variability (HRV change, last 5 min – first 5 min). The demographic variables of age and sex were also included in all moderation models to control for their potential effects.

**Nonparametric tests.** For all DVs collected at both timepoints (pre, post), changes over time were tested with nonparametric t-tests using Wilcoxon signed-rank tests in JASP [68]. Effect sizes were reported as the matched rank biserial correlation (RBC), an appropriate measure for nonparametric t-tests which ranges between –1 to 1 and indicates the relative proportion of positive and negative rank scores while ignoring ties (e.g., RBC = 0.60 means that 60% more subjects showed an increase instead of a decrease). Timepoint tests were performed separately for each of the four experimental groups (MindGym (Breathwork), MindGym (Rain), VR (Breathwork), VR (Rain)) as well as the combined MindGym and VR groups, the combined Breathwork and Rain groups, and all groups combined.

For all DVs, differences between groups were tested with nonparametric Mann-Whitney U tests in JASP, which included RBC effect sizes. Six separate group comparisons were tested: MindGym (Breathwork) vs MindGym (Rain), VR (Breathwork) vs VR (Rain), MindGym (Breathwork) vs VR (Breathwork), MindGym (Rain) vs VR (Rain), MindGym vs VR, and Rain vs Breathwork. For any DV collected at both timepoints, the difference score (Post – Pre) was used to compare group differences in changes over time. For the physiological measures, the first timepoint, last timepoint, and the difference between first and last timepoints were separately used as DVs to compare group differences in initial, final, or change in physiological response.

**Moderation.** Moderation of the experimental effects on mood or cognition was tested as interactions between each covariate and the Tech and Stim group factors in separate models for the three-way interaction (Tech x Stim x Moderator) and two-way interactions (Tech x Moderator, Stim x Moderator), as well as a separate model with only main effects. Demographic covariates (age and sex) were also included as main effects in all models. For any outcome measure collected at both timepoints, the difference score (Post – Pre) was used as the DV. Because we did not have any a priori hypotheses about differential moderation effects between nested levels of Tech and Stim, we conducted post-hoc t-tests for only the interaction terms that were deemed significant.

Moderations were analyzed with generalized linear models (GLM), using the *glmmTMB* [69] and *emtrends* [70] packages in RStudio [71]. Based on extensive model diagnostics, which were conducted with the *DHARMa* [72] package, it was determined that either gaussian or t-family distributions (both with the identity link) were optimal for model fitting to ensure no substantial violations or problems with linearity, independence of errors, homoscedasticity, dispersion, zero-inflation, outliers, or within-group normality of residuals.

Separate models were tested for the hypothesized moderators, which included eight psychological variables (DPES, Immersion, IPQ General, IPQ Involvement, IPQ Spatial Presence, IPQ Experienced Realism, MODTAS, and Openness) and seven physiological variables (overall motion, initial Breath Rate, Breath Rate change, initial HR, HR change, initial HRV, HRV change). All moderation models were tested separately for 10 outcome variables as DVs: Arousal, Anxiety (STAI), Awe, BSE, EDI, Total Mood Disturbance (POMS), Valence, Architex (Total Score), Architex (Total Speed), and TMT (RTACC).

**Repeated measures ANOVA.** For the physiological measures collected during the experiences, repeated measures ANOVAs (rmANOVA) were performed in JASP to assess changes over time as a main effect of Timepoint (two timepoints for HRV, five timepoints for Breath Rate and EDA, ten timepoints for HR) and to assess experimental effects on changes over time, with separate three-way (Tech x Stim x Timepoint) or two-way interactions (Tech x Timepoint, Stim x Timepoint). Levene's test was used to ensure no violations of homogeneity. The assumption of sphericity was tested (using Mauchly's test) and corrected (with Greenhouse-Geisser) if necessary. Effect sizes for all main or interaction effects were reported as generalized eta-squared, which appropriately adjusts for mixed design with within-subject (Time) and between-subjects (Tech, Stim) factors. Effect sizes for post hoc t-tests, if an interaction was significant, were reported as Cohen's *d.*

**Follow up analyses.** Three outcome variables - STAI, POMS (Total Mood Disturbance), and a discrete version (10 total responses ranging from 'not at all anxious' to 'extremely anxious') of the generalized anxiety visual analog scale (GA-DS) – were collected at the 1-week follow-up timepoint. For the two variables (STAI and POMS Total Mood Disturbance) collected at all three timepoints (pre, post, and follow-up), rmANOVAs were performed in JASP to assess changes over time as a main effect of Timepoint (pre, post, follow-up) and to assess experimental effects on changes over time, with separate three-way (Tech x Stim x Timepoint) or two-way interactions (Tech x Timepoint, Stim x Timepoint). One variable (GA-DS) was

collected only at the follow-up timepoint, so an ANOVA was performed to compare groups. Levene's test was used to ensure no violations of homogeneity. The assumption of sphericity (for models including Timepoint) was tested (using Mauchly's test) and corrected (with Greenhouse-Geisser) if necessary. Effect sizes for all main or interaction effects were reported as generalized eta-squared (for models including Timepoint) or partial eta-squared (for models without Timepoint). Effect sizes for any relevant post hoc t-tests were reported as Cohen's *d*.

**Multiple comparison correction.** We used the false discovery rate (FDR) method of correcting *P* values for multiple comparisons (significance threshold chosen as $a$=.05, $P$<.05) to control Type I and Type II errors [73,74]. FDR correction was performed across DVs and separately for 1) the nonparametric tests of timepoint effects for each group (single or combined), 2) the nonparametric tests of group differences, separately for each group comparison and separately for the measures collected at both timepoints, the measures collected at only the Post timepoint, and the physiological measures collected during the experience, as well as 3) the moderation tests, separately for each interaction or main effect. FDR correction of the rmANOVAs was performed across the different models testing interaction or main effects and separately for each measure.

**Mediation.** Mediation analysis was conducted in JASP based on the *lavaan* [75] package in R. Missing values were excluded listwise. Separate models were tested for each potential mediator, which included the same variables that were used as potential moderators in the moderation analysis, and for each DV that was also used in the moderation analysis.

## Neurophysiological analysis

**Data acquisition and preprocessing.** Physiological data, including electrocardiogram (ECG), electrodermal activity (EDA), electromyography (EMG), and respiration, were recorded using the CGX AIM Phys. Mon. AIM-0106 device. The data were stored in XDF format, which is a standardized format for multi-channel time series data [76]. The XDF files were loaded using the pyxdf library, and the relevant data streams were extracted based on their names and labels.

**ECG processing.** The ECG data were first sanitized to remove any invalid or missing data points [77]. A notch filter was then applied to remove powerline interference at 60 Hz. The ECG signal was further cleaned by applying a bandpass filter with a low cut-off frequency of 0.5 Hz and a high cut-off frequency of 45 Hz using a 4th-order Butterworth filter [78]. The filtered ECG data were then filtered again to remove any values exceeding a predefined amplitude threshold of 1500 mV.

Baseline wandering, a low-frequency artifact in the ECG signal, was computed by measuring the peak-to-peak amplitude of the ECG signal. If the baseline wandering exceeded a predefined threshold of 1700 mV, the ECG data were considered unacceptable for further analysis.

**EDA processing.** The EDA data were processed in conjunction with the EMG data to identify and remove motion artifacts. First, the EMG signal was normalized. A sliding window approach was then used to identify windows of EMG data that contained a high proportion of outliers, which were considered to be indicative of motion artifacts. The Local Outlier Factor (LOF) algorithm [79] was used to detect outliers within each window (implemented using the scikit-learn library) [80].

The EDA signal was smoothed using a median filter with a kernel size of 8 seconds. A low-pass Butterworth filter with a cut-off frequency of 5 Hz was then applied to remove high-frequency noise. The contaminated windows identified in the EMG signal were used to guide the interpolation of the EDA signal. Linear interpolation was performed to replace the artifact-contaminated segments of the EDA signal with values interpolated from the neighboring clean segments.

The quality of the cleaned EDA signal was assessed by checking for values outside the acceptable range (0.05 - 60 µS) and for rapid changes exceeding ±10 µS/sec within a 1-second interval [81]. The proportion of data points violating these criteria was used to compute a

quality index ranging from 0 to 100, with higher values indicating better signal quality. If the quality index was below 50, the EDA data were considered unacceptable for further analysis.

**Respiration processing.** The respiration data were preprocessed following the methods of [82]. We first applied a high-pass Butterworth filter with a cut-off frequency of 0.1 Hz to remove low-frequency drift. The filtered signal was then resampled to a fixed sampling rate of 100 Hz. A sliding window approach was used to estimate the respiration period within each window using the Fast Fourier Transform (FFT). The estimated respiration period was used to guide the application of a moving average filter to smooth the respiration signal within each window. The smoothed signal was then resampled back to the original sampling rate.

Inhalation and exhalation onsets were detected by identifying the intercepts between the smoothed respiration signal and its moving mean. The intercepts were then classified as inhalation or exhalation onsets based on the direction of the slope at each intercept. A minimum amplitude threshold of 0.25 was used to filter out small fluctuations that do not correspond to true respiratory events.

The number of breaths within the first and last 5 minutes of the experiment was computed using the inhalation and exhalation onsets to define individual breaths.

**Heart rate variability analysis.** Heart rate variability (HRV) analysis was performed by computing the low-frequency (LF) and high-frequency (HF) components of the ECG signal using the Welch's method for power spectral density estimation [83]. The LF component was defined as the power in the frequency range of 0.04-0.15 Hz, while the HF component was defined as the power in the range of 0.15-0.4 Hz. The ratio of LF to HF power (LF/HF ratio) was computed as a measure of sympathovagal balance [84]. The LF/HF ratio was computed separately for the first and last 5 minutes of the experiment.

**EDA tonic component analysis.** The tonic component of the EDA signal, which reflects the slow-varying level of skin conductance, was extracted by applying a low-pass filter to the EDA signal [77]. The mean value of the tonic component was computed within non-overlapping 2-minute intervals. These mean values were used as a measure of the overall level of arousal throughout the experiment.

## Ethics

**IRB.** The Advarra (Columbia, MD) Institutional Review Board approved all recruitment and testing procedures before initiating enrollment (Pro00070581). Following ethical standards and legal requirements, all study participants provided written informed consent on a document, hosted through DropboxSign. They were afforded ample time to seek clarification from the Principal Investigator and study personnel. The informed consent process incorporated the California Experimental Research Bill of Rights as mandated by Health and Safety Code Section 24172. Additionally, the study adhered to the principles outlined in the Declaration of Helsinki. Furthermore, all laboratory personnel possessed and maintained up-to-date certifications in Good Clinical Practice and the Protection of Human Research Participants through online training.

## Results

### Pre-post changes as a function of group

**Timepoint effects within group.** Analysis of pre-post changes revealed significant improvements across various measures when considering all groups combined and within specific conditions (see Table 4 and S1 Tables A-I for detailed results).

When considering all groups combined, there was evidence of a decrease in TMT (RTACC) ($W$=5443.00, $Z$=4.32, $p.fdr$<.001, $RBC$=.45), Architex (Total Speed) ($W$=6044.00,

Z=7.82, *p.fdr*<.001, *RBC*=.84), Anxiety (STAI) (*W*=5156.50, *Z*=4.86, *p.fdr*<.001, *RBC* =.52), Valence (*W*=1626.00, *Z*=-2.14, *p.fdr*=.042, *RBC*=-.26), and POMS (*W*=5659.00, *Z*=5.77, *p.fdr*<.001, *RBC*=.61).

Examining specific conditions, the MindGym (Breathwork) group demonstrated improvements in Architex (Total Speed) (*W*=396.00, *Z*=3.36, *p.fdr*<.001, *RBC*=.70) and POMS (*W*=331.00, *Z*=2.92, *p.fdr*=.014, *RBC*=.63). The MindGym (Rain) condition showed enhancements for TMT (RTACC) (*W*=411.00, *Z*=3.19, *p.fdr*<.001, *RBC*=.66), Architex (Total Speed) (*W*=421.00, *Z*=4.40, *p.fdr*<.001, *RBC*=.94), STAI (*W*=353.50, *Z*=2.94, *p.fdr*=.005, *RBC*=.63), and POMS (*W*=442.00, *Z*=3.80, *p.fdr*<.001, *RBC*=.78).

The VR (Breathwork) group exhibited improvements in Architex (Total Speed) (*W*=355.00, *Z*=3.99, *p.fdr*<.001, *RBC*=.88), STAI (*W*=405.00, *Z*=3.55, *p.fdr*<.001, *RBC*=.74), and POMS (*W*=406.00, *Z*=3.10, *p.fdr*=.005, *RBC*=.64) (see S1 Table C), while the VR (Rain) condition only showed a significant improvement in Architex (Total Speed) (*W*=379.00, Z=4.01, *p.fdr*<.001, RBC=.87).

When comparing the combined Breathwork and Rain conditions, both showed significant improvements across multiple measures, with slight variations in effect sizes. Similarly, the combined MindGym and VR conditions significantly enhanced cognitive performance, anxiety reduction, and mood improvement.

**Group differences in timepoint effects.** Analysis of group differences in timepoint effects revealed no statistically significant differences between experimental conditions after FDR correction for multiple comparisons (all *p.fdr*>.05) (see S2 Tables A-F). While all interventions yielded positive outcomes across various measures, including improvements in cognitive performance, anxiety levels, and mood, the magnitude of these changes did not differ

**Table 4. Summary of results from the nonparametric (Wilcoxon Signed-Rank) t-tests of timepoint (pre vs post) differences for the different groups or group combinations.**

| DV | MindGym (Breathwork) | | | MindGym (Rain) | | | VR (Breathwork) | | | VR (Rain) | | |
|---|---|---|---|---|---|---|---|---|---|---|---|---|
| | ΔM | RBC | sig | ΔM | RBC | sig | ΔM | RBC | sig | ΔM | RBC | sig |
| TMT (RTACC) | 2.68 | 0.45 | ~ | -42.64 | 0.66 | *** | -43.21 | 0.33 | | -35.42 | 0.33 | |
| Architex (Total Speed) | -51.77 | 0.70 | *** | -72.88 | 0.94 | *** | -114.97 | 0.88 | *** | -69.26 | 0.87 | *** |
| Architex (Total Score) | -0.93 | 0.37 | | 0.65 | -0.34 | | -0.37 | 0.25 | | -0.96 | 0.22 | |
| Anxiety (STAI) | -3.68 | 0.49 | ~ | -4.09 | 0.63 | ** | -5.70 | 0.74 | *** | -1.43 | 0.16 | |
| Arousal | -0.19 | 0.13 | | -0.38 | 0.11 | | 0.70 | -0.29 | | -0.63 | 0.28 | |
| Valence | 1.03 | -0.51 | ~ | 0.41 | -0.45 | ~ | -0.12 | -0.07 | | 0.03 | -0.03 | |
| POMS | -9.90 | 0.63 | * | -10.50 | 0.78 | *** | -10.58 | 0.64 | ** | -7.17 | 0.31 | |

| DV | Breathwork | | | Rain | | | MindGym | | | VR | | | All | | |
|---|---|---|---|---|---|---|---|---|---|---|---|---|---|---|---|
| | ΔM | RBC | sig | ΔM | RBC | sig | ΔM | RBC | sig | ΔM | RBC | sig | ΔM | RBC | sig |
| TMT (RTACC) | -21.6 | 0.391 | * | -39.1 | 0.51 | *** | -20.35 | 0.56 | *** | -39.9 | 0.33 | * | -30.4 | 0.45 | *** |
| Architex (Total Speed) | -81.7 | 0.791 | *** | -71.1 | 0.9 | *** | -62.33 | 0.82 | *** | -91.7 | 0.87 | *** | -76.4 | 0.84 | *** |
| Architex (Total Score) | -0.67 | 0.312 | ~ | -0.13 | -0 | | -0.141 | 0.06 | | -0.67 | 0.23 | | -0.4 | 0.14 | |
| Anxiety (STAI) | -4.72 | 0.621 | *** | -2.81 | 0.4 | * | -3.889 | 0.55 | *** | -3.67 | 0.47 | ** | -3.78 | 0.52 | *** |
| Arousal | 0.266 | -0.12 | | -0.5 | 0.22 | | -0.285 | 0.12 | | 0.063 | 0.01 | | -0.11 | 0.06 | |
| Valence | 0.437 | -0.26 | | 0.226 | -0.2 | | 0.715 | -0.5 | ** | -0.05 | -0.1 | | 0.333 | -0.3 | * |
| POMS | -10.3 | 0.636 | *** | -8.89 | 0.59 | *** | -10.21 | 0.71 | *** | -8.95 | 0.5 | *** | -9.58 | 0.61 | *** |

'DV' refers to the dependent variable. 'ΔM' refers to the mean difference (post – pre). 'RBC' refers to rank biserial correlation as a measure of effect size (ranging from -1 to 1) and reflects the relative proportions of positive or negative ranks. 'sig' refers to the FDR-corrected *P*-value where '~' indicates.05<*p.fdr*<.10, '*' indicates *p.fdr*<.05, '**' indicates *p.fdr*<.01, and '***' indicates *p.fdr*<.001.

significantly between experimental conditions when accounting for multiple comparisons. This lack of significant between-group differences indicates that no single condition demonstrated superior efficacy over the others in terms of the measured variables.

**Outcome measures as a function of group.** No statistically significant differences were observed between the group comparisons for the post-intervention measures (all $p.fdr>.05$) (see S3 Tables A-F). The closest result to reaching statistical significance with a notable effect size was found for IPQ (Involvement) when comparing Breathwork to Rain ($W=2501.50$, $p.fdr=.12$, $RBC=.26$), suggesting a potential trend towards higher involvement in the Breathwork condition compared to the Rain condition.

## Physiology measures

**Group differences.** Analysis of physiological measures revealed significant differences between experimental conditions, particularly in breath rate patterns, the most notable being between the Breathwork and Rain conditions (see S4 Tables A-E for all results).

At the end of the session, Breathwork participants had significantly lower final Breath Rate compared to Rain ($W=269.50$, $p.fdr<.001$, $RBC=-.70$). The overall change in Breath Rate from beginning to end of the experience was significantly different between the two conditions ($W=198.50$, $p.fdr<.001$, $RBC=-.78$), indicating a more substantial decrease in breath rate for the Breathwork group.

Similar patterns were observed when comparing the MindGym (Breathwork) and Mind-Gym (Rain) conditions. MindGym (Breathwork) had a significantly lower final Breath Rate ($W =77.00$, $p.fdr<.001$, $RBC=-.68$) compared to MindGym (Rain), and the overall change in Breath Rate showed greater decrease, with a significantly different Breath Rate ($W=68.50$, $p.fdr<.001$, $RBC=-.72$) with MindGym (Breathwork).

VR (Breathwork) and VR (Rain) showed significant differences in breath rate patterns, with significantly lower Breath Rate at the end of the experience ($W=55.00$, $p.fdr<.001$, $RBC=-.73$) compared to VR (Rain). The overall change in Breath Rate was significantly different ($W=37.00$, $p.fdr<.001$, $RBC= -.82$), with VR (Breathwork) demonstrating a more substantial decrease.

These results consistently demonstrate that participants in the Breathwork conditions, regardless of the delivery method (MindGym or VR), experienced a decrease in breath rate throughout the session compared to those in the Rain conditions. This pattern suggests that the Breathwork interventions effectively modulated participants' breathing patterns, potentially leading to a more relaxed physiological state by the end of the session.

No other physiological measures (EDA, HR, HRV) showed significant differences between conditions after FDR correction.

**Changes over time.** We employed repeated measures ANOVA (rmANOVA) to examine changes over time and the potential effects of experimental conditions. We observed significant temporal changes in HR, Breath Rate, and EDA levels, with only Breath Rate showing interactions between time and technology. No significant main effects of time or interactions with groups were observed for HRV after FDR correction.

The results revealed a significant main effect of time on HR ($F_{5.17, 398.38}=8.78$, $p.fdr=.004$), indicating a gradual increase throughout the experiment (see Table 5). EDA levels revealed a significant main effect of time ($F_{2.35, 162.37}=17.11$, $p.fdr=.004$), characterized by an initial increase followed by a plateau around the third timepoint.

Breath Rate showed a significant main effect of time ($F_{2.02, 171.33}=25.31$, $p.fdr=.002$), characterized by an initial decrease followed by a plateau around the third timepoint. A significant interaction between time and Stimuli was also observed ($F_{2.48, 208.49}=28.49$, $p.fdr=.002$). Post hoc

**Table 5. Results from repeated measures ANOVA on physio measures.**

| | Timepoint | | | | | | Timepoint x Tech | | | | | |
|---|---|---|---|---|---|---|---|---|---|---|---|---|
| DV | df1 | df2 | F | p.raw | p.fdr | $\eta^2_G$ | df1 | df2 | F | p.raw | p.fdr | $\eta^2_G$ |
| HR | 5.17 | 398.38 | 8.78 | <.001 | 0.004 | 0.005 | 5.14 | 390.59 | 0.56 | 0.738 | 0.777 | 0.003 |
| HRV | 1.00 | 77.00 | 1.54 | 0.219 | 0.438 | 0.009 | 1.00 | 76.00 | 0.31 | 0.581 | 0.615 | 0.002 |
| EDA | 2.35 | 162.37 | 17.11 | <.001 | 0.004 | 0.020 | 2.35 | 159.51 | 0.92 | 0.414 | 0.552 | 0.001 |
| BR | 2.02 | 171.33 | 25.31 | <.001 | 0.002 | 0.130 | 2.02 | 169.50 | 0.38 | 0.685 | 0.867 | 0.002 |
| | Timepoint x Stim | | | | | | Timepoint x Tech x Stim | | | | | |
| DV | df1 | df2 | F | p.raw | p.fdr | $\eta^2_G$ | df1 | df2 | F | p.raw | p.fdr | $\eta^2_G$ |
| HR | 5.16 | 392.00 | 0.84 | 0.527 | 0.777 | 0.005 | 5.09 | 376.86 | 0.50 | 0.777 | 0.777 | 0.003 |
| HRV | 1.00 | 76.00 | 2.67 | 0.106 | 0.424 | 0.015 | 1.00 | 74.00 | 0.26 | 0.615 | 0.615 | 0.002 |
| EDA | 2.37 | 161.14 | 1.90 | 0.146 | 0.292 | 0.003 | 2.36 | 155.43 | 0.18 | 0.871 | 0.871 | 0.002 |
| BR | 2.48 | 208.49 | 28.49 | <.001 | 0.002 | 0.140 | 2.48 | 203.01 | 0.19 | 0.867 | 0.867 | 0.001 |

DV: dependent variable; HR = heart rate; HRV = heart rate variability; EDA = electrodermal activity (tonic); BR = breath rate. Timepoint = main effect model (within-subjects); Timepoint x Tech = interaction model (between-groups: MindGym vs VR); Timepoint x Stim = interaction model (between-groups: Breathwork vs Rain); Timepoint x Tech x Stim = three-way interaction model. F = F-test with df1, df2 (degrees of freedom); P.RAW/P.FDR = uncorrected/FDR-corrected p-values.

paired-sample t-tests (with Holm correction of *P* values) were conducted between timepoints within Stimuli groups. Within the Rain groups combined, breath rate did not reliably change between any timepoints (all *p.holm*>.05). Within the Breathwork groups combined, breath rate significantly decreased from the first to second timepoint (Δ*M*=-3.55, *SE*=.49, *t*=7.19, *p.holm*<.001, *d*=1.08), remained similar between second and third timepoints (Δ*M*=-1.12, *SE*=0.49, *t*=2.26, *p.holm*=.416, *d*=.034), decreased again from third to fourth timepoints (Δ*M*=-2.28, *SE*=.49, *t* = 4.62, *p.holm* <.001, *d*=.69), without any further reliable decrease from fourth to fifth timepoints (Δ*M*=.77, *SE*=.49, *t*=1.56, *p.holm*=1.000, *d*=.23).

**Mediation.** Mediation analyses were conducted to test if hypothesized variables mediated the group effects on the outcome scores. No mediation analyses passed the threshold for significance after FDR correction (all *p.fdr*>.05).

**Moderation.** Our moderation analysis revealed significant effects of both psychological and physiological factors on participants' experiences and performance during the immersive interventions (see S5 Tables A-J for detailed results). Among the psychological trait measures, MODTAS emerged as a significant predictor of both Awe (*b*=.37, *SE*=.11, *Z*=3.28, *p.fdr*=.004) (see Table 6) and EDI (*b*=.03, *SE*=.01, *Z*=2.98, *p.fdr*=.015), with higher MODTAS scores associated with greater reported Awe and EDI. DPES also showed a significant main effect on Awe (*b*=13.89, *SE*=3.48, *Z*=4.00, *p.fdr*<.001), indicating that individuals with higher dispositional positive emotions were more likely to experience awe during the intervention.

Psychological state measures also played a significant role in moderating participants' experiences. Immersion showed strong main effects on both Awe (*b*=1.16, *SE*=0.22, *Z*=5.17, *p.fdr*<.001) and EDI (*b*=0.07, *SE*=0.02, *Z*=4.55, *p.fdr*<.001), with higher levels of immersion consistently associated with greater reported awe and emotional depth. IPQ (General) was also positively associated with Awe (*b*=6.18, *SE*=1.72, *Z*=3.59, *p.fdr*=.002) and EDI (*b*=0.32, *SE*=0.13, *Z*=2.49, *p.fdr*=.048). Only IPQ (Involvement) followed this same pattern with Awe (*b*=2.24, *SE*=0.76, *Z*=2.96, *p.fdr*=.009) and EDI (*b*=0.18, *SE*=0.05, *Z*=3.24, *p.fdr*=.009).

Turning to physiological moderators, we found the change in heart rate throughout the intervention (HR-diff) showed a significant interaction effect with the type of technology used on Architex (Total Score) ($F_{df1}$=10, *p.fdr*=.023). Specifically, an increase in heart rate was associated with decreased accuracy for participants in the MindGym groups, but not for those in the VR.

**Table 6. Results from GLM tests of moderation.**

| DV | Tech x Stim x Moderator | | | Tech x Moderator | | | Stim x Moderator | | | Moderator | | | | |
|---|---|---|---|---|---|---|---|---|---|---|---|---|---|---|
| Awe | F | p.raw | p.fdr | F | p.raw | p.fdr | F | p.raw | p.fdr | b | SE | Z | p.raw | p.fdr |
| **Psych Moderator** | | | | | | | | | | | | | | |
| DPES | 1.74 | 0.190 | 0.824 | 0.14 | 0.711 | 0.887 | 0.20 | 0.657 | 0.957 | 13.89 | 3.48 | 4.00 | <.0001 | <.001 |
| Immersion | 0.03 | 0.870 | 0.892 | 0.08 | 0.776 | 0.887 | 0.02 | 0.896 | 0.957 | 1.16 | 0.22 | 5.17 | <.0001 | <.001 |
| IPQ (General) | 0.02 | 0.892 | 0.892 | 4.70 | 0.032 | 0.362 | 1.92 | 0.169 | 0.853 | 6.18 | 1.72 | 3.59 | <.001 | 0.002 |
| IPQ (Involvement) | 1.08 | 0.301 | 0.892 | 0.12 | 0.727 | 0.887 | 0.82 | 0.367 | 0.853 | 2.24 | 0.76 | 2.96 | 0.003 | 0.009 |
| IPQ (Spatial Presence) | 0.38 | 0.538 | 0.892 | 0.00 | 0.957 | 0.957 | 0.77 | 0.381 | 0.853 | 1.83 | 0.93 | 1.98 | 0.048 | 0.104 |
| IPQ (Experienced Realism) | 1.52 | 0.220 | 0.824 | 0.34 | 0.560 | 0.887 | 3.98 | 0.048 | 0.725 | 0.68 | 0.87 | 0.78 | 0.436 | 0.620 |
| MODTAS | 0.56 | 0.458 | 0.892 | 0.05 | 0.828 | 0.887 | 0.38 | 0.539 | 0.899 | 0.37 | 0.11 | 3.28 | 0.001 | 0.004 |
| Openness | 0.36 | 0.553 | 0.892 | 0.85 | 0.359 | 0.796 | 0.03 | 0.871 | 0.957 | 1.22 | 0.81 | 1.51 | 0.131 | 0.218 |
| **Physio Moderator** | | | | | | | | | | | | | | |
| Motion (overall) | 2.69 | 0.104 | 0.824 | 0.14 | 0.711 | 0.887 | 1.41 | 0.238 | 0.853 | 0.79 | 0.50 | 1.59 | 0.112 | 0.210 |
| Breath Rate (first 2 min) | 0.13 | 0.724 | 0.892 | 1.08 | 0.301 | 0.796 | 0.06 | 0.805 | 0.957 | 0.15 | 1.02 | 0.15 | 0.883 | 0.883 |
| HR (first 1 min) | 2.45 | 0.122 | 0.824 | 0.59 | 0.446 | 0.837 | 0.01 | 0.908 | 0.957 | -0.70 | 0.29 | -2.39 | 0.017 | 0.042 |
| HRV (first 5 min) | 0.03 | 0.853 | 0.892 | 4.04 | 0.048 | 0.362 | 0.00 | 0.957 | 0.957 | 42.30 | 65.70 | 0.64 | 0.520 | 0.650 |
| Breath Rate change (last - first) | 0.02 | 0.876 | 0.892 | 0.81 | 0.371 | 0.796 | 0.82 | 0.367 | 0.853 | -0.19 | 0.82 | -0.24 | 0.814 | 0.872 |
| HR change (last - first) | 0.17 | 0.686 | 0.892 | 2.44 | 0.123 | 0.508 | 0.72 | 0.398 | 0.853 | -0.60 | 0.80 | -0.75 | 0.455 | 0.620 |
| HRV change (last - first) | 0.68 | 0.412 | 0.892 | 2.28 | 0.135 | 0.508 | 0.55 | 0.461 | 0.865 | 9.92 | 38.69 | 0.26 | 0.798 | 0.872 |

DV (dependent variable) = Awe (post timepoint); Psych Moderator/Physio Moderator = psychological/physiological moderating variable; Tech x Stim x Moderator = three-way interaction model (moderator × Tech [MindGym vs VR] × Stimuli [Breathwork vs Rain]); Tech x Moderator = two-way interaction (moderator × Tech); Stim x Moderator = two-way interaction (moderator × Stimuli); Moderator = main effects model; F = F-test (GLM); b/SE = unstandardized beta coefficient/standard error; Z = Z-test; p.raw/p.fdr = uncorrected/FDR-corrected p-values.The trait of Openness demonstrated an interaction effect with the type of stimuli on anxiety levels as measured by STAI ($F_{1,111}$=9.58, *p.fdr*=.038). Specifically, higher levels of openness were associated with a greater reduction in anxiety for participants in Breathwork, but not for those in Rain.

## Descriptives

Young adults (18-34) showed the highest motion sickness susceptibility (mean = 20.58, SD = 13.74) but the lowest gastric symptoms (mean = 12.95, SD = 5.20), suggesting their intervention response may have been influenced by motion-related discomfort (see Table 7). Older adults (55+) experienced the highest levels of gastric (mean = 16.30, SD = 15.74), central (mean = 18.30, SD = 13.43), and peripheral symptoms (mean = 18.52, SD = 15.74). Males displayed higher central symptom scores across all ages and more consistent peripheral symptoms, while females showed greater symptom variability across all categories. Sopite symptoms (drowsiness and fatigue) were consistent across age groups (mean ≈ 23.0), with higher scores among younger males.

**Follow up analyses.** Follow-up analysis results are fully reported in the Supplemental Materials (S6 Tables A-C; S6 Fig). Both STAI and POMS (Total Mood Disturbance) showed significant changes over time, without any interactions between groups, such that scores significantly decreased immediately after the experience (consistent with the nonparametric results reported earlier) but then significantly increased one week later to a similar level (POMS) or even higher level (STAI) compared to before the experience. Effect sizes of pairwise group comparisons were small to moderate (ranging from 0.09 to 0.68). For the measure of anxiety (GA-DS) collected only at the 1-week timepoint, there were no significant

Table 7. Motion sickness scores across demographics and groups.

| Motion Sickness | | Total Score | | Gastrointestinal | | Central | | Peripheral | | Sopite-related | |
|---|---|---|---|---|---|---|---|---|---|---|---|
| | | M | SD | M | SD | M | SD | M | SD | M | SD |
| **Age** | | | | | | | | | | | |
| | 18-34 | 20.58 | 13.74 | 17.19 | 14.63 | 20.38 | 15.03 | 18.24 | 16.43 | 26.00 | 17.87 |
| | 35-54 | 15.73 | 4.80 | 12.47 | 4.63 | 15.45 | 5.96 | 14.13 | 7.15 | 20.54 | 9.93 |
| | 55+ | 15.02 | 3.55 | 11.57 | 2.55 | 14.67 | 5.27 | 11.73 | 1.71 | 21.39 | 9.57 |
| **Gender** | | | | | | | | | | | |
| | Female | 17.85 | 9.36 | 13.84 | 8.52 | 17.35 | 10.22 | 15.46 | 12.33 | 24.25 | 14.78 |
| | Male | 17.36 | 10.26 | 14.64 | 11.71 | 17.32 | 11.65 | 15.11 | 11.18 | 21.83 | 13.01 |
| **Group** | | | | | | | | | | | |
| | Cube | 18.39 | 10.37 | 15.43 | 11.84 | 18.52 | 12.42 | 16.40 | 13.63 | 22.66 | 11.47 |
| | VR | 16.82 | 9.17 | 13.05 | 8.19 | 16.16 | 9.11 | 14.17 | 9.42 | 23.41 | 16.09 |
| | Breath | 17.87 | 10.62 | 14.45 | 9.94 | 17.71 | 12.23 | 16.72 | 12.98 | 22.35 | 14.03 |
| | Rain | 17.33 | 8.92 | 14.02 | 10.56 | 16.95 | 9.46 | 13.80 | 10.15 | 23.75 | 13.88 |
| | MindGym (Breathwork) | 19.00 | 10.18 | 15.23 | 9.13 | 19.50 | 13.95 | 17.32 | 13.57 | 23.39 | 12.08 |
| | MindGym (Rain) | 17.80 | 10.69 | 15.63 | 14.12 | 17.57 | 10.89 | 15.51 | 13.84 | 21.96 | 10.98 |
| | VR (Breathwork) | 16.81 | 11.08 | 13.72 | 10.73 | 16.03 | 10.30 | 16.16 | 12.59 | 21.38 | 15.77 |
| | VR (Rain) | 16.83 | 6.68 | 12.31 | 3.91 | 16.30 | 7.77 | 11.98 | 2.32 | 25.65 | 16.40 |

Age groups are divided into 18-34, 35-54, and 55+ years. M and SD represent the mean and standard deviation.

main effects of Tech groups ($F_{(1,85)}$ = 0.31, p = 0.577, partial $\eta^2$ = 0.004), Stim groups ($F_{(1,85)}$ = 0.11, p = 0.738, partial $\eta^2$ = 0.001), or interaction between Tech and Stimuli groups ($F_{(1,83)}$ = 0.88, p = 0.352, partial $\eta^2$ = 0.010).

## Discussion

This study aimed to investigate the impact of content delivery systems on two different anxiolytic stimuli: guided breathwork and naturalistic rain. We compared the efficacy of a novel "reflective chamber" (MindGym) with a traditional virtual reality (VR) headset in delivering these interventions. While our initial hypothesis anticipated clear superiority of one technology over the other, our results instead demonstrated the robust effectiveness of MindGym-generated content across both delivery systems. This finding highlights the MindGym's potential as a powerful content generation platform, capable of producing experiences that maintain their efficacy even when translated to more portable VR formats.

Our findings revealed significant reductions in anxiety measures across all conditions, with the STAI showing significant improvements in all groups except for the breathwork condition in the MindGym. These results may be of clinical relevance and are comparable to benchmarks from other acute anxiety interventions. Our study demonstrated an overall STAI reduction of 11.67% across all groups, with breathwork showing the highest reduction at 15.10%. These findings align with recent studies on non-pharmacological interventions for acute anxiety reduction (e.g., Johnson et al. [4] reported a 15% reduction in STAI scores in non-anxious populations exposed to a psychedelic-esque stroboscopic lights and binaural beats experience, and a 13% reduction in response to breath-focused meditation; Sriboonlert et al. [85] found a 17% reduction in anxiety scores following aromatherapy interventions; et al. [86] observed a 7% reduction from exercise in lower trait anxious individuals and a 17% reduction for those with higher trait anxiety).

Our results suggest that both the MindGym and VR delivery systems, combined with either rain or breathwork stimuli, can produce anxiety reductions comparable to these established interventions. The fact that such basic interventions, which do not fully showcase the complex experiences unique to the MindGym, were equally effective in both systems highlights the MindGym's potential as a content creation tool. It suggests that even more sophisticated MindGym-generated experiences could be successfully adapted for VR, potentially enhancing their accessibility without compromising efficacy. However, it's important to note that these effects are still considerably lower than the 30-50% reduction typically observed with pharmacological interventions [87]. This comparison highlights the potential for non-pharmacological approaches while also acknowledging the current limitations in their efficacy compared to medication-based treatments. Interestingly, we observed no significant changes in arousal levels, while valence only improved significantly in the MindGym condition. The interpretation of pre-post results centered around task performance (e.g., Architex) is challenging without a control condition, as some improvements may be attributable to practice effects. For instance, the TMT typically shows improvement with repeated administrations [88,89].

Contrary to our initial expectations, we found no main effects of either stimulus type or technology on the outcome measures. This lack of differentiation between the MindGym and VR conditions is particularly noteworthy, as it suggests that the MindGym-generated content retained its effectiveness when translated to a VR format. While there were numerical trends suggesting potentially better affective outcomes in the MindGym (e.g., greater reduction in STAI scores and mood disturbance), these differences did not reach statistical significance. We observed a consistent pattern of "more" significant effects in the MindGym across most measures, but we caution against over-interpreting these trends. Surprisingly, we did not observe the expected differences in immersion between the two technologies. This lack of differentiation may suggest that both systems were equally effective in creating an immersive experience, or that our measures were not sensitive enough to detect subtle differences. It also points to the successful adaptation of MindGym-generated content for VR platforms.

The most notable physiological finding was the decrease in breath rate in the breathwork condition compared to the rain condition. This result serves as a sanity check, confirming that participants engaged with the breathwork instructions as intended. However, we found no significant differences in electrodermal activity (EDA), heart rate (HR), or heart rate variability (HRV) as a function of technology or stimulus type.

We explored whether individual differences could account for variations in response to the interventions. Several interesting moderation effects emerged. DPES predicted the amount of awe experienced, though this finding is somewhat expected given the overlap between DPES and awe measures. MODTAS, which has previously been shown to predict audiovisual-induced chills [90], had a main effect on both awe and ego dissolution. MODTAS also moderated the effects of stimuli on anxiety reduction, with higher trait absorption associated with greater anxiety reduction in the breathwork condition across both technologies. Openness to experience moderated the impact on STAI scores, with more open individuals experiencing greater anxiety reduction in the breathwork condition, but not in the rain condition.

These moderation effects exclusive to breathwork may reflect the importance of explicit instructions and participants' willingness to fully engage with the experience. The breathwork condition may have benefited individuals who are more prone to absorption and open to new experiences, possibly due to its more structured nature compared to the passive rain condition. Immersion and presence, while not differing between conditions, were potent predictors of awe and ego dissolution across all groups. This aligns with previous research showing that the degree of immersion in virtual environments impacts memory encoding and neural

representations [41]. Interestingly, motion sickness negatively impacted ego dissolution, suggesting that physical discomfort may anchor individuals in their bodily sensations, hindering the experience of ego transcendence.

Contrary to our expectations, young adults (18-34) showed higher motion sickness susceptibility (mean = 20.58, SD = 13.74) than middle-aged (mean = 15.73, SD = 4.80) and older adults (mean = 15.02, SD = 3.55). Additionally, MindGym conditions unexpectedly showed slightly higher motion sickness scores (Breathwork: mean = 19.00, Rain: mean = 17.80) compared to VR conditions (Breathwork: mean = 16.81, Rain: mean = 16.83). While this appears to contradict established theories about vestibular mismatch in VR environments (Reggente et al., 2018), it's important to note that our use of 360° video rather than fully interactive VR may have created a different type of perceptual experience than typically studied in VR research, as the non-interactive nature of both conditions could have affected motion sickness in ways not predicted by traditional vestibular mismatch theory. Males and females showed similar total motion sickness scores (males: mean = 17.36, females: mean = 17.85), though females exhibited greater variability across symptom categories. Sopite symptoms remained relatively consistent across age groups (mean ≈ 23.0), with both conditions showing similar levels (MindGym: mean = 22.66, VR: mean = 23.41), with a slight advantage to the MindGym, making it potentially more ideal for reliably inducing mood states.

Study retention patterns revealed differences between participants who completed follow-up assessments and those who did not, particularly across age groups and intervention conditions. The most notable difference was age, with survey completers being about 8 years older on average (43.6 vs 35.6 years), while other demographic differences were relatively modest in comparison. Another group difference was also observed in intervention type, with MindGym conditions showing notably higher completion rates (76% combined) compared to VR conditions (58% combined).

In conclusion, this study contributes to the growing body of research on non-pharmacological interventions for acute anxiety reduction while also demonstrating the potential of the MindGym as a versatile content creation platform. The comparable efficacy of both the MindGym and VR systems, using content originally generated for the MindGym, suggests that experiences designed in the MindGym can be successfully translated to more portable VR formats without significant loss of effectiveness. This finding is particularly promising given that the interventions used in this study were relatively basic and did not fully leverage the unique capabilities of the MindGym. Future research should focus on developing more complex, immersive experiences using the MindGym's full potential, and exploring how these can be effectively adapted for VR and other delivery systems. This approach could lead to the development of highly effective, accessible, and scalable interventions for anxiety management, potentially bridging the gap with pharmacological treatments while offering the benefits of non-pharmacological approaches.

## Limitations

A significant limitation of this study is that the "VR" condition was not a true virtual reality experience, but rather a 3D video displayed in a VR headset. A more equitable comparison of the content delivery systems would have involved creating a virtual environment that mimicked the MindGym chamber and responded more dynamically head movements, potentially enhancing immersion and presence and serving as a VR-analog as opposed to a recording of the MindGym experience. The current design may have inadvertently positioned the MindGym experience as a "steel man" argument, with many of its immersive properties carried over to the VR condition.

However, this limitation also highlights the potency of the MindGym programming, demonstrating that its effects can be successfully replicated using a VR device. This finding suggests that the content displayed in a refelctie chamber environment may be the primary driver of the observed anxiolytic effects that extend into VR.

The unexpected age and platform differences in motion sickness susceptibility challenge conventional assumptions about cybersickness in younger populations and digital environments and may warrant the development of new scales for non-VR-based experiential tehcnologies. The substantial individual variability in symptoms, particularly among younger adults and across platforms, represents a study limitation and emphasizes the need for personalized comfort settings and adaptive monitoring systems in future research designs. Additionally, future studies should consider implementing both fully interactive VR environments and more detailed measures of cybersickness and environmental satisfaction to better understand how these factors might influence intervention effectiveness. Such modifications would help distinguish intervention effectiveness from cybersickness effects while accounting for unexpected susceptibility patterns across different types of digital experiences.

Our follow-up analyses revealed that while the interventions produced significant immediate reductions in anxiety and mood disturbance, these beneficial effects were not maintained one week post-intervention. Particularly noteworthy was the observation that anxiety levels not only returned to baseline but exceeded pre-intervention measurements at follow-up, suggesting a potential rebound effect that warrants further investigation. These findings indicate that while single-session interventions can provide acute anxiety relief, maintaining long-term benefits may require regular practice or multiple sessions, especially given the apparent vulnerability to anxiety rebound effects. However, it's important to note that follow-up measurements were taken outside the controlled laboratory environment where pre- and post-intervention assessments occurred, potentially introducing different contextual factors and stressors. Furthermore, without also having measurements at one week *prior* to the start of the intervention (which would mirror the one week *post* intervention), it's uncertain how these follow-up levels compare to subjects' average mood states in general, irrespective of their participation in the experiment. This could be an interesting question for future studies.

## Future directions

Future research should focus on conducting experiments with more visually diverse content to identify where these content delivery systems meaningfully diverge. This could involve developing true VR environments that more closely mimic the MindGym experience, allowing for a more direct comparison of the technologies. Investigating the impact of these delivery systems on different types of anxiety (e.g., social anxiety, specific phobias) could determine if certain technologies are more effective for particular anxiety subtypes. Exploring the long-term effects of repeated exposures to these interventions would assess their potential as ongoing anxiety management tools. Examining the role of individual differences (e.g., absorption, openness) in treatment response could lead to the development of more personalized anxiety interventions.

The lack of significant differences in these physiological measures across conditions may suggest that the anxiolytic effects are mediated through cognitive or emotional pathways rather than direct physiological changes. Therefore, investigating the neural correlates of the anxiolytic effects observed in this study would better our understanding of the mechanisms underlying the effectiveness of these interventions. Additionally, this might account for some individual differences in responses that were not seen by self-reported trait variables.

Future research should explore the potential utility of brief digital relaxation interventions as pre-session tools in clinical settings. Given the potent, acute anxiolytic effects observed in our study, these interventions could help standardize patients' arousal states before therapy sessions, potentially offsetting situational stressors (e.g., commute-related anxiety) that might otherwise impact session effectiveness. Such application could be particularly valuable in urban clinical settings where external stressors are common. Investigating whether these interventions can create a more consistent baseline state for therapeutic engagement would be an important next step in evaluating their ecological utility in clinical practice.

## Conclusion

In conclusion, while our study did not find significant differences between the MindGym and VR delivery systems, it demonstrated that the powerful anxiolytic effects originally discovered in the reflective chamber environment successfully translated to cirtual reality without loss of efficacy. The observed moderation effects highlight the importance of individual differences in treatment response and suggest that tailoring interventions to personal characteristics may enhance their effectiveness. This successful virtual adaptation of an established anxiety-reducing environment opens promising avenues for developing more accessible and effective anxiety management tools.

## Supporting information

**S1 Table.** A. MindGym Breathwork. B. MindGym Rain. C. VR Breathwork. D. VR Rain. E. MindGym. F. VR. G. Breathwork. H. Rain. I. All.
(PDF)

**S2 Table.** A. MindGym Breathwork vs MindGym Rain. B. MindGym Breathwork vs VR Breathwork. C. MindGym Rain vs VR Rain. D. VR Breathwork vs VR Rain. E. Breathwork vs Rain. F. MindGym vs VR.
(PDF)

**S3 Table.** A. MindGym Breathwork vs MindGym Rain. B. MindGym Breathwork vs VR Breathwork. C. MindGym Rain vs VR Rain. D. VR Breathwork vs VR Rain. E. Breathwork vs Rain. F. MindGym vs VR.
(PDF)

**S4 Table.** A. MindGym Breathwork vs MindGym Rain. B. MindGym Breathwork vs VR Breathwork. C. MindGym Rain vs VR Rain. D. VR Breathwork vs VR Rain. E. Breathwork vs Rain. F. MindGym vs VR.
(PDF)

**S5 Table.** A. Architex Total Score. B. Architex Total Speed. C. TMT TRACC. D. Arousal. E. POMS. F. STAI. G. Valence. H. Awe. I. EDI. J. BSE.
(PDF)

**S6 Table.** A. Descriptive Statistics. B. Repeated measures ANOVA. C. Post hoc comparisons.
(PDF)

**S6 Fig. Descriptive Statistics by Timepoints.**
(TIFF)

**S1 Appendix. Lumena Labs Infinity Cube Spec Sheet.**
(PDF)

**S2 Appendix. Transcription of 4-8 Breathwork Audio.**
(PDF)

## Author contributions

**Conceptualization:** Ninette Simonian, Nicco Reggente.

**Data curation:** Ninette Simonian, Nicco Reggente.

**Formal analysis:** Ninette Simonian, Micah Alan Johnson, Velu Kumaravel, Taylor Kuhn, Felix Schoeller, Nicco Reggente.

**Funding acquisition:** Nicco Reggente.

**Investigation:** Ninette Simonian, Caitlin Lynch, Geena Wang, Nicco Reggente.

**Methodology:** Ninette Simonian, Micah Alan Johnson, Nicco Reggente.

**Project administration:** Ninette Simonian, Nicco Reggente.

**Resources:** Nicco Reggente.

**Supervision:** Nicco Reggente.

**Validation:** Ninette Simonian, Nicco Reggente.

**Visualization:** Ninette Simonian.

**Writing – original draft:** Ninette Simonian, Micah Alan Johnson, Caitlin Lynch, Geena Wang, Velu Kumaravel, Taylor Kuhn, Felix Schoeller, Nicco Reggente.

**Writing – review & editing:** Ninette Simonian, Micah Alan Johnson, Caitlin Lynch, Geena Wang, Velu Kumaravel, Taylor Kuhn, Felix Schoeller, Nicco Reggente.

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
