## [Decision Letter · Decision Letter 0]

22 Nov 2024

PMEN-D-24-00465

Contrasting cognitive, behavioral, and physiological  responses to breathwork vs. naturalistic stimuli in reflective chamber and VR headset environments

PLOS Mental Health

Dear Dr. Simonian,

Thank you for submitting your manuscript to PLOS Mental Health. After careful consideration, we feel that it has merit but does not fully meet PLOS Mental Health’s publication criteria as it currently stands. Therefore, we invite you to submit a revised version of the manuscript that addresses the points raised during the review process.

The manuscript has been evaluated by two reviewers, and their comments are available below.

The reviewers have raised a number of major concerns. They request revisions to the data presentation and further consideration to potential limitations of the study, such as sample size and generalisability.

Could you please carefully revise the manuscript to address all comments raised?

We look forward to receiving your revised manuscript.

Kind regards,

Helen Howard

Staff Editor

PLOS Mental Health

Journal Requirements:

1. We ask that a manuscript source file is provided at Revision. Please upload your manuscript file as a .doc, .docx, .rtf or .tex.

Additional Editor Comments (if provided):

Reviewers' comments:

Reviewer's Responses to Questions

**Comments to the Author**

1. Does this manuscript meet PLOS Mental Health’s publication criteria ? Is the manuscript technically sound, and do the data support the conclusions? The manuscript must describe methodologically and ethically rigorous research with conclusions that are appropriately drawn based on the data presented.

Reviewer #1: Yes

Reviewer #2: Yes

2. Has the statistical analysis been performed appropriately and rigorously?

Reviewer #1: No

Reviewer #2: Yes

3. Have the authors made all data underlying the findings in their manuscript fully available (please refer to the Data Availability Statement at the start of the manuscript PDF file)?

Reviewer #1: Yes

Reviewer #2: Yes

4. Is the manuscript presented in an intelligible fashion and written in standard English?

Reviewer #1: Yes

Reviewer #2: Yes

5. Review Comments to the Author

Reviewer #1: Dear Author,

Thank you for your submission. To enhance the clarity and comprehensiveness of your data presentation, I recommend adding the following tables:

Repeated Measures ANOVA Table: Please include a table that presents the results of the repeated measures ANOVA. This table should detail the within-subjects and between-subjects effects, along with relevant F-values, degrees of freedom, p-values, and effect sizes.

Follow-Up Data Table: A table displaying follow-up data would help readers assess the sustainability of the intervention's effects or any time-related changes. Please include key descriptive and inferential statistics for each follow-up period.

Dropout Data Table: Providing a table that outlines dropout rates and reasons for attrition will add transparency to the study's methodology. Including demographics of those who dropped out versus those who completed the study could also be helpful.

These additions will strengthen the presentation of your findings and offer readers a more detailed understanding of your results.

Best regards,

Ranjit Singha

Reviewer #2: In my perspective, the study addresses an innovative and timely area in mental health research, exploring the comparative efficacy of VR and physical immersive environments for including anxiety reduction, mood enhancement, and cognitive function improvement. Combining VR and a physical reflective chamber with two distinct stimuli (rain and breathwork) and examining their effect on cognitive and physiological outcomes adds to the novelty. Particularly, the use of individual differences like MODTAS and Openness to explain responses is a novel approach to personalize treatment, which may be of significant interest in precision mental health interventions.

The methodology appears robust, using a randomized assignment of 126 participants to four conditions, which allows for control over confounding variables and increases confidence in the study's findings. The use of both physiological and psychological measures (cognitive tests, breath rate, and anxiety scales) offers a well-rounded assessment of the interventions' impact. However, it might be worth noting a few points for potential improvement, for example sample size and generalizability: although 126 participants is a decent sample size, understanding if the sample is representative (e.g., age, gender, socioeconomic status, previous experience with VR, mental health status…) would help assess the generalizability of the findings. Moreover, assessing the presence of cybersickness or satisfaction with the digital environments could provide valuable insights, as these factors may have influenced participants' responses and overall effectiveness of the interventions. These points could also be a limitation and not only the use of a 360º video instead of a VR condition to compare to MindGym.

It would be interesting to see the study discuss practical implications for mental health professionals, particularly those working with clients with anxiety, as this could be highly relevant to readers of PLOS Mental Health.

Line 741: authors should not use contractions “weren’t seen”, please change to “were not see”.

6. PLOS authors have the option to publish the peer review history of their article (what does this mean? ). If published, this will include your full peer review and any attached files.

**Do you want your identity to be public for this peer review?** For information about this choice, including consent withdrawal, please see our Privacy Policy .

Reviewer #1: **Yes: ** Ranjit Singha

Reviewer #2: No

---

## [Decision Letter · Decision Letter 1]

3 Feb 2025

Contrasting cognitive, behavioral, and physiological  responses to breathwork vs. naturalistic stimuli in reflective chamber and VR headset environments

PMEN-D-24-00465R1

Dear Ms. Simonian,

We are pleased to inform you that your manuscript 'Contrasting cognitive, behavioral, and physiological  responses to breathwork vs. naturalistic stimuli in reflective chamber and VR headset environments' has been provisionally accepted for publication in PLOS Mental Health.

Best regards,

Juan Felipe Cardona, Ph.D.

Academic Editor

PLOS Mental Health

Reviewer Comments (if any, and for reference):

Reviewer's Responses to Questions

**Comments to the Author**

1. If the authors have adequately addressed your comments raised in a previous round of review and you feel that this manuscript is now acceptable for publication, you may indicate that here to bypass the “Comments to the Author” section, enter your conflict of interest statement in the “Confidential to Editor” section, and submit your "Accept" recommendation.

Reviewer #2: All comments have been addressed

Reviewer #3: All comments have been addressed

2. Does this manuscript meet PLOS Mental Health’s publication criteria ? Is the manuscript technically sound, and do the data support the conclusions? The manuscript must describe methodologically and ethically rigorous research with conclusions that are appropriately drawn based on the data presented.

Reviewer #2: Yes

Reviewer #3: Yes

3. Has the statistical analysis been performed appropriately and rigorously?

Reviewer #2: Yes

Reviewer #3: Yes

4. Have the authors made all data underlying the findings in their manuscript fully available (please refer to the Data Availability Statement at the start of the manuscript PDF file)?

Reviewer #2: (No Response)

Reviewer #3: Yes

5. Is the manuscript presented in an intelligible fashion and written in standard English?

Reviewer #2: Yes

Reviewer #3: Yes

6. Review Comments to the Author

Reviewer #2: (No Response)

Reviewer #3: The article presents an innovative proposal for addressing symptoms of anxiety and stress. Although the sample size may be considered small, the study utilizes various sources of information, including behavioral and physiological measures. The presentation and analysis of these measures are appropriate. While the use of multiple measurements may be somewhat confusing, they are adequately presented, and the conclusions are consistent with the recorded data and the limitations of the study are appropriately considered.

All comments were addressed in the revision.

7. PLOS authors have the option to publish the peer review history of their article (what does this mean? ). If published, this will include your full peer review and any attached files.

**Do you want your identity to be public for this peer review?** For information about this choice, including consent withdrawal, please see our Privacy Policy .

Reviewer #2: No

Reviewer #3: No
